# Genome-Wide Scan of Wool Production Traits in Akkaraman Sheep

**DOI:** 10.3390/genes14030713

**Published:** 2023-03-14

**Authors:** Yunus Arzik, Mehmet Kizilaslan, Sedat Behrem, Stephen N. White, Lindsay M. W. Piel, Mehmet Ulas Cinar

**Affiliations:** 1Department of Animal Science, Faculty of Agriculture, Erciyes University, 38039 Kayseri, Türkiye; 2International Center for Livestock Research and Training Center, Ministry of Agriculture and Forestry, 06852 Ankara, Türkiye; 3Department of Animal Science, Faculty of Veterinary Medicine, Aksaray University, 68100 Aksaray, Türkiye; 4Department of Veterinary Microbiology & Pathology, College of Veterinary Medicine, Washington State University, Pullman, WA 99164, USA; 5USDA-ARS Animal Disease Research 3003 ADBF, Washington State University, Pullman, WA 99164, USA

**Keywords:** heritability, QTL, GWAS, wool traits, sheep

## Abstract

The objective of this study was to uncover the genetic background of wool quality, a production trait, by estimating genomic heritability and implementing GWAS in Akkaraman sheep. The wool characteristics measured included fibre diameter (FD) and staple length (SL) at the age of 8 months and yearling fibre diameter (YFD), yearling staple length (YSL) and yearling greasy fleece weight (YGFW) at 18 months of age. Animals were genotyped using the Axiom 50 K Ovine Genotyping Array. Maximum likelihood estimations of a linear mixed model (LMM) were used to estimate genomic heritability, where GWAS was conducted following a score test of each trait. Genomic heritability estimates for the traits ranged between 0.22 and 0.63, indicating that phenotypes have a moderate range of heritability. One genome- and six chromosome-wide significant SNPs were associated with the wool traits in Akkaraman lambs. Accordingly, *TRIM2*, *MND1*, *TLR2*, *RNF175*, *CEP290*, *TMTC3*, *RERE*, *SLC45A1*, *SOX2*, *MORN1*, *SKI*, *FAAP20*, *PRKCZ*, *GABRD*, *CFAP74*, *CALML6* and *TMEM52* genes as well as nine uncharacterized regions (*LOC101118971*, *LOC105609137*, *LOC105603067*, *LOC101122892*, *LOC106991694*, *LOC106991467*, *LOC106991455*, *LOC105616534* and *LOC105609719*) were defined as plausible candidates. The findings of this study shed light on the genetics of wool quality and yield for the Akkaraman breed and suggests targets for breeders during systematic breeding programmes.

## 1. Introduction

Small ruminant breeding has played an important role in meeting human needs (meat, milk, leather and wool) for centuries [1]. Sheep breeding creates economic value through wool production by producing approximately 1160 million kilograms of clean wool in the world each year [2]. Small ruminant breeding brings people food and job security, with strength in the ability to adapt these animals to environmental extremes [3]. Turkey is among the top 10 countries for domestic sheep producers considering the large number of breeds and the approximately 45 million sheep housed and grown [4].

Akkaraman is a fat-tailed dual purpose (meat and milk) breed, and it is the most commonly raised indigenous sheep in Türkiye [5,6]. Populations of the breed have not been subjected to systematic genetic selection to increase fleece quality, leaving the wool characteristics relatively low, especially in terms of fineness, compared with fine-wool sheep such as the Merino. Several studies suggested that Akkaraman sheep have a coarse- and carpet-type wool [7,8]. This lower quality fleece is generally used for manufacturing carpets, rugs and blankets, whereas the clothing industry uses high quality wool, which results in increased value [9,10]. Various studies have shown that an individual-level of variation exists and affects fleece quality parameters, emphasising the role the genetic background plays on the traits of interest [11].

The economic value of fleece is directly related to its fineness, length, strength, fibre elasticity and final weight. Within the clothing textile industry, lower diameter and strong resistance are preferred, as they prevent breaks which may occur throughout the production process. Wool quality parameters are mainly affected by the genetic background of the animals and environmental factors such as nutrition, climate and age [12]. Studies have been carried out in alternate breeds of sheep to estimate the heritability of wool quality traits and to uncover genomic regions that affect wool traits. Prior work has identified moderate heritability estimates for fleece traits. Within studied yearling sheep, it was reported that heritability for fibre diameter ranged between 0.16 and 0.74, where heritability estimates of staple length were found to be between 0.33 and 0.76. Lastly, greasy fleece weight at first shearing had a heritability estimate reported between 0.30 and 0.77 [2,13,14,15,16,17,18].

In the last two decades, genome research in farm animals has been accelerated by the advances in molecular genetics. Reference genomes have been published for domestic sheep and high density commercial arrays have been developed for genome-wide distributed single nucleotide polymorphisms (SNPs), which can be employed to map the relationship between economically important traits and genomic loci [19]. Genomic data have been used effectively in many studies to discover SNPs, major variants, genes and quantitative trait loci (QTLs) associated with traits in farm animals [20,21,22,23,24,25]. The information obtained from these studies can be incorporated into systematic breeding programmes to achieve rapid genetic improvement of the studied traits [26,27]. Along with other economically important traits in sheep, genome-wide association studies (GWAS) have been carried out to uncover the genetic background of wool traits [28]. In GWAS studies conducted on fine-wool sheep breeds, various genes and families (i.e., *KRTCAP3*, *KRTAP9*, KAP family, *UBE2E3* and *RHPN2*) predominantly involved in keratin synthesis were suggested to play a role in fibre diameter, staple length and greasy fleece weight, representing the most economically important wool parameters [2,29,30]. While prior studies exist, they remain extremely rare, hampering continuous progress towards revealing genetic regions behind these complex traits. Therefore, the objective of this study was to uncover the genetic background of wool traits in Akkaraman sheep, which is the most popular indigenous sheep breed in Türkiye. For this purpose, trait heritability was estimated based on available genomic data and GWAS were implemented to investigate possible associated genomic regions for wool fibre and fleece characteristics.

## 2. Materials and Methods

### 2.1. Animals and Phenotypes

In total, 426 Akkaraman sheep (Appendix A) raised in three commercial farms around the outskirts of Ankara, Türkiye, were used in the study. Fleece samples were taken from all 426 (141 males, 285 females) animals at the age of 8 months. At 18 months of age, approximately 78 of these female animals had a second fleece sampling, which was additionally weighed for greasy fleece (an example of Akkaraman wool presented as Appendix A). Further information on the climatic and geographic conditions of the region, herd management, mating, selection strategy and feeding practices were presented in detail previously [22].

Analysis of fibre diameter and staple length required approximately 50 g of fleece sampled from the caudal aspect of the scapula. These were taken at 8 and 18 months of age and delivered to the laboratory in numbered sample bags. Samples sent to the laboratory were analysed in terms of fibre diameter and length using OFDA 2000 (BSC Electronics, Ardross, Australia), which is an optical fibre diameter analyser instrument located in the Ankara Sheep and Goat Breeders Association Laboratory. During the analysis, a bunch of fibres were aligned and straightened from end to end and placed into the device. Following loading, fibres were automatically measured optically in micrometres (µm) for diameter and millimetres (mm) for length. At 18 months of age, the animals were sheared in a clean area and the resultant wool weighed using a scale with a precision of 10 g. In total, five wool traits were measured and recorded as following: fibre diameter at 8 months of age (FD), staple length at 8 months of age (SL), yearling fibre diameter (YFD), yearling staple length (YSL) and yearling greasy fleece weight (YGFW). The descriptive statistics of the traits are presented in Table 1.

### 2.2. Genotype and Quality Control

Genotyping was performed using the Axiom 50 K Ovine Genotyping Array (Thermo Fisher Sci, Waltham, MA, USA) at the Genetics Laboratory of the International Centre for Livestock Research and Training (ICLRT) in Ankara. During genotyping, the Axiom 2.0 Assay 96 Array Format Manual Workflow protocols provided by the manufacturer were followed. Quality control (QC) of the raw genomic data was performed using the GenABEL package in the R programme [31]. Detailed information on genotyping steps and QC analysis criteria are also presented by [20]. Briefly, SNPs with a call rate below 95%, minor allele frequency lower than 0.05 and which were out of Hardy–Weinberg Equilibrium (*p* = 0.05/49,931) were removed from the analysis. Additionally, those SNPs that are mapped on the sex-linked chromosomes, mitochondrial chromosomes as well as the SNPs with unknown positions were also removed from further analysis. On the other hand, animals were removed from the analysis if samples had a call rate below 90% and a pairwise IBS rate higher than 95%.

### 2.3. Statistical Analysis

Missing genotypes were imputed before proceeding to genomic heritability estimation and genome-wide association analyses. SNPs that passed the QC criteria were imposed and imputed according to the current status of the SNP within the given population. The expectation-maximisation (EM) algorithm was used with the process carried out by the R GenABEL package [31].

The animal linear mixed model and average information restricted maximum likelihood (AI-REML) approach provided by the ‘sommer’ package of R [32] was used to equate genomic heritability estimations. Significant environmental factors were included in the models as a fixed effect and a genomic relationship matrix was utilised to capture the covariance between individuals in terms of the phenotypes of interest. The animal mixed model is given below:y=Xβ+Zu+e
where ***y*** is the vector of individual observations of each trait measured, ***β*** is the vector of fixed environmental effects (sex, birth type, birth month, etc.), u is the polygenic background effect obtained from MVN (***u***~0, $G\sigma_{u}^{2}$), e is the vector of random residual errors obtained from MVN (***e***~0, $I\sigma_{e}^{2}$), ***X*** is the design matrices mapping fixed effects and ***Z*** is the polygenic background effects of each observation. Within MVN calculation, $\sigma_{u}^{2}$ and $\sigma_{e}^{2}$ are the additive genetic variance and random residual variances for a particular trait, respectively. Furthermore, ***I*** is the identity matrix and ***G*** represents the genomic relationship matrix derived by VanRaden [33]. The following is the formula for the G matrix:$$G=\frac{ZZ^{\prime }}{2\sum p_{i}\left(1-p_{i}\right)}$$

After estimating variance components for each trait, the narrow sense heritability estimates of each trait were obtained by using the formula given below:$$h^{2}=\frac{\sigma_{u}^{2}}{\sigma_{p}^{2}\left(\sigma_{u}^{2}+\sigma_{e}^{2}\right)}$$
where $h^{2}\;$ is the heritability estimate of each trait, $\sigma_{u}^{2}$ is the estimate of genetic variance for each trait, $\sigma_{p}^{2}$ is the total phenotypic variance and $\sigma_{e}^{2}$ is the error variance. The environmental factors are included in the model as a fixed effect.

The same generalised model and genomic relationship matrix were used to estimate SNP effects in genome-wide association studies with a score test developed by Chen and Abecasis [34]. In the model, the additive effects of SNPs were estimated by employing a genomic relationship matrix (i.e., SNP-based identity-by-state information) to account for covariance due to cryptic relatedness between animals and population stratification. A widely used graphical content, the Manhattan plot (Figure 1), was used to visualise the −log10 (*p*-value) of the SNP effects for a sense of global genomic distribution. Q-Q plots of the SNPs and genomic inflation factor lambda (λ) were also utilised to observe potential inflation of the test statistics. Genomic controls were applied to the *p*-values of all SNPs in the analysis of each trait to avoid any potential inflation after the use of the mixed model analysis approach [35]. A conservative threshold, determined with Bonferroni correction, for statistical significance was used to avoid an increased type 1 error rate (i.e., probability of rejection if the null hypothesis is true) due to the multiple comparisons. Consequently, the genome-wide threshold was defined as 1.18 × 10^−6^ (0.05/42,320), with the chromosome-wide threshold as 3.07 × 10^−5^ ((0.05/42,320) × 26).

### 2.4. Functional Annotation of Significant SNPs

Positional information for SNPs associated with measured traits was ascertained using the Oar_v4.0 genome assembly and the NCBI Genome Data Viewer [36]. Candidate genes were defined as genes which overlapped with SNPs. If the established SNP was not located on a gene, the nearest gene was suggested as a candidate. DAVID Bioinformatics Resources 2021 was used to obtain biological information on the identified candidate genes [37]. Additionally, previously identified quantitative trait loci (QTL) related to wool traits were checked through the Animal QTL Database to see if there was any overlap with the SNPs identified in this study [38]. In case of insufficient annotation in the sheep reference genome, orthology between species was used. Moreover, the annotation of specific genes from cattle, goats, mice and humans were evaluated for shared biological function. Finally, the metabolic pathways involving candidate genes were inspected by employing their Gene Ontology terms, which can be explored further on the QuickGo website [39].

## 3. Results

### 3.1. Descriptive Statistics of Phenotypic and Genotypic Data

Preceding further analysis, phenotypes were checked for outliers. Measurements that were below or above three standard deviations were considered outliers and excluded from further analysis. The phenotypic data were also checked for normality, where only yearling fibre diameter (YFD) was found to slightly deviate from a normal distribution, which was transformed to provide a normal distribution. The log transformation applied to YFD prior to any further analysis basically changed the scale of the data into log_10_ base, which removed the skewedness of the data and established a near normal distribution. The resultant statistical analysis of the phenotypes are presented in Table 1. The mean fibre diameter (FD) and staple length at 8 months of age were 23.17 µm and 22.73 mm, respectively. Alternatively, at 18 months of age the yearling fibre diameter (YFD) and yearling staple length (YSL) were 34.50 µm and 73.91 mm, respectively. Lastly, the mean greasy fleece weight (YGFW) was found to be 2.43 kg. Detailed information on the descriptive statistics of the traits (i.e., mean, minimum, maximum and coefficient of variation) and corresponding numbers of the traits are present in Table 1.

Genotyping resulted in 49,931 SNPs called for the sample set. These raw data were filtered against various QC standards and 42,320 SNPs for each sample passed the imposed criteria. Therefore, further analyses were based on these 42,320 SNPs from varying animal numbers (see Table 1), which depended upon the trait of interest.

### 3.2. Genetic Parameter Estimates

Statistical power was increased following quality control by imputation of genomic data for missing SNPs before genomic relationship matrix construction and association analyses. Expected genotype scores for imputation of missing SNPs were estimated based on the population mean as described by Chen and Abecasis [34]. An animal mixed model was used to estimate genetic parameters, while a genomic relationship matrix was obtained from the genomic data to estimate additive genetic variance.

The simple linear model was used to determine the effects of fixed factors before preceding to genetic parameter estimations and GWAS analysis. Accordingly, herd and age (in days) were found to be significant on fibre diameter and staple length. Meanwhile, greasy fleece weight was not significantly affected by fixed factors. Genetic parameter estimates (i.e., additive genetic variances, residual variances and genomic heritability estimations) for traits and their standard errors are illustrated in Table 2. Heritability of traits ranged from 0.22 for YFD to 0.63 for YSL. This suggests that the heritability of measured traits ranges from moderate to relatively high. However, these heritability measures are low overall compared with those calculated for fine-wool sheep breeds [2,13,40,41,42].

### 3.3. Genome-Wide Association Studies

GWAS results correlating measured wool production traits with significant SNPs are presented in Table 3. This table also contains found associations corresponding to *p*-values, associated genes and genome- or chromosome-wide significance. Type one errors may be imposed through multiple SNP testing; to avoid these errors, both genome- and chromosome-wide thresholds were calculated. Thresholds were determined using the Bonferroni correction test [43]. Accordingly, the genome-wide significance threshold was set to 1.18 × 10^−6^ (0.05/42,320) and the chromosome-wide significance threshold was set to 3.07 × 10^−5^ ((0.05/42,320) × 26). Lambda (λ) was calculated to check for the presence of inflation or deflation in the corrected *p*-values for SNPs. A visualisation of expected versus observed *p*-values are presented in Figure 2 as quantile–quantile (Q-Q) plots.

A total of seven SNPs, one genome- and six chromosome-wide, on sheep chromosomes (OAR) 1, 2, 3, 8, 12 and 17 were found to be significantly associated with the analysed traits. Notably, a significant SNP was not identified for the yearling greasy fibre weight (YGFW) phenotypic trait. Genes associated with significant SNPs were located by searching the Oar_v4.0 sheep genome assembly either 200 kb upstream or downstream of the significant loci through the use of the National Centre for Biotechnology Information (NCBI) Genome Data Viewer [36]. Accordingly, the significant SNPs were related to 17 candidate genes, namely *TRIM2*, *MND1*, *TLR2*, *RNF175*, *CEP290*, *TMTC3*, *RERE*, *SLC45A1*, *SOX2*, *MORN1*, *SKI*, *FAAP20*, *PRKCZ*, *GABRD*, *CFAP74*, *CALML6* and *TMEM52.* Besides these defined genes, there were nine undefined regions isolated, namely *LOC101118971*, *LOC105609137*, *LOC105603067*, *LOC101122892*, *LOC106991694*, *LOC106991467*, *LOC106991455*, *LOC105616534* and *LOC105609719*, as candidates. Importantly, two of the chromosome-wide SNPs did not associate with any defined or undefined genes.

## 4. Discussion

Akkaraman wool has long been known to have coarse- and carpet-type wool. Most of the wool produced by Akkaraman sheep has been used for blanket, rug and carpet production since a couple of historical studies suggested that Akkaraman sheep have low quality wool [7,8]. However, these findings were based only on a few studies implemented decades ago. Therefore, no known selection was applied on the wool characteristics of Akkaraman sheep. However, in our study, we found that the fibre diameter of Akkaraman wool is not that far from those mainstream sheep that are known for fine-wool production for the clothing industry, such as Merino. Therefore, with this study, we unravelled overall wool characteristics of Akkaraman as comparable with those mainstream fine-wool sheep and investigated the genetic background of those wool characteristics in Akkaraman sheep, so that, in need of a genetic selection programme for wool characteristics, enhanced methods such as marker-assisted selection could be utilised for faster genetic improvement.

### 4.1. Genetic Parameter Estimates

In farm animals, most of the economically important traits are measured as a quantitative character. Importantly, quantitative characters tend to have complex structures because they are controlled by a large number of genes dispersed within the genome, the heritable component of variation. Therefore, it is essential to identify and understand the genetic parameters (i.e., heritability) responsible for altered animal characteristics to employ sustainable genetic improvement breeding programs. Notably, pedigree records are needed to calculate genetic parameter estimates of economically important traits and are significantly lacking for indigenous sheep breeds. Fortunately, deep pedigree records are not required to calculate present genomic relationships, which allowed the present study to estimate the genomic heritability of wool production traits. The five chosen traits within this study were fibre diameter (FD), staple length (SL), yearling fibre diameter (YFD), yearling staple length (YSL) and yearling greasy fleece weight (YGFW), which implemented animal mixed models using GRM to calculate the genetic parameters of interest.

Fibre diameter heritability estimates were found to be 0.41 and 0.22 at 8 and 18 months, respectively. These values are quite low compared with the renowned fine-wool sheep breeds. In particular, prior studies on Chinese fine-wool sheep suggested a heritability of yearling fibre diameter of 0.64 [2]. Moreover, heritability for yearling fibre diameter has been reported from 0.65 to 0.75 in the world-renowned Merino and its crossbreeds [40,41,42]. Heritability estimates for fibre diameter in Targhee and Polypay sheep breeds were 0.41 and 0.36, respectively, close to the estimates obtained in our study [15,16]. On the other hand, a study conducted in Rambouillet sheep determined a low heritability estimate of 0.16 [14].

A second desired trait is a long staple length, where the increased length results in a higher yield during the industrial processing of the wool. Studies considering fine-wool sheep indicate that the heritability for staple length (SL) at first shearing varies between 0.23 and 0.66 in Merino and between 0.44 and 0.62 in Chinese fine-wool sheep [2,40,41,42]. Furthermore, other sheep breeds (i.e., Columbia, Rambouillet, Targhee, Polypay and Menz sheep), were found to have heritabilities which also widely varied from 0.33 to 0.76 for SL and YSL [13,14,15,16,17]. With these widely ranging values, it can be presumed that staple length is heavily affected by both the physiological condition of the animal and environmental factors.

The heritability estimate of yearling greasy fleece weight in Merino and its crosses has been found between 0.32 and 0.53 [40,41,42]. This value has remained consistent within other sheep breeds, which ranged from 0.30 to 0.68 [13,14,15,16,17,18]. Therefore, the value obtained within the present study (0.48) generally agrees with values reported in the literature. One exception to this is the high heritability of yearling greasy fleece weight defined for Chinese fine-wool breeds (between 0.67 and 0.77) [2,44].

There may be two reasons for the high variation of heritability for fibre diameter and length at the different ages represented in our study population. Firstly, the data consist of the different sample sizes, 410 animals at 8 months of age and 78 animals at 18 months of age, and alternate sampling time frames. Another reason for high heritability estimate variance could simply be the physiological differences between the two stages of life, where the animal genetics or environment might be more effective in the determination of the trait at young versus adult periods. Nevertheless, it is clear that many factors (i.e., the genetic architecture of the traits, population structure of breeds, the analytical model and the scientific approach) play a role in the differences of heritability estimates in studies conducted in different sheep breeds.

### 4.2. Genome-Wide Association Studies (GWAS)

Within the last two decades, GWAS has become a widely used method to uncover genetic mechanisms behind phenotypic expression and to identify genomic factors (i.e., SNPs, QTLs, genes and genomic regions) that act on economically important traits in farm animals. Species-specific commercial SNP panels containing functional variants that span the whole genome are a relatively inexpensive and quick methodology used to obtain genomic information from animals. In our study, GWAS were carried out to reveal the genomic regions that affect the production and quality parameters of wool. An animal-mixed-model-based association test and genomic control were implemented to avoid any possible false positives. Overall, the used methods appear to be successful considering the Q–Q plots of all the analysed traits, as there is no major systematic inflation of the test statistics. The special case of the YFD and YSL (i.e., the slight deflation of the test statistics) might be sourced by the low sample size, as well as the very highly polygenic nature of the traits, which is also supported by the non-association of any major SNPs regarding the traits of interest [43].

As seen in Table 3, one genome-wide and two chromosome-wide SNPs were detected for the fibre diameter (FD) measured at 8 months. *rs399017576* was the SNP variant with the highest *p*-value (1.03 × 10^−7^) detected on OAR17 and was the only genome-wide significant SNP. This SNP is localised on an uncharacterised genomic region (*LOC101122892*). The most significant chromosome-wide SNP (*rs416693666*) was found on OAR3, near the centrosome protein 290 (*CEP290*) gene. This gene encodes a protein which localises to centrosomes and cilia. Annotated biological functions for *CEP290* included cilium assembly (GO: 0060271), ciliary basal body-plasma membrane docking (GO: 0097711) and non-motile cilium assembly (GO: 1905515). Cilia are critical for hair follicle development. A study conducted on mice found that cilia are a key signalling component for normal hair morphogenesis and is needed for cells in the dermis to receive signals from hedgehog signalling pathways [45]. The preponderance of data showing ciliary importance for hair fibre characteristics suggests that the *CEP290* gene is a strong candidate for further analyses. During the association analysis for the yearling fibre diameter trait (YFD), a separate SNP was found to have chromosome-wide significance on OAR2; however, this SNP was not associated with any gene.

The study of staple length at 8 months of age returned significance at SNPs *rs399379059* and *rs420575132* on OAR1 and 12, respectively. *rs399379059* is present within the arginine-glutamic acid dipeptide repeats (*RERE*) gene. This gene encodes an atropine family of arginine-glutamic acid dipeptide repeat-containing proteins. Meanwhile, the significant SNP associated with the staple length measured at 18 months (YSL), *rs419440256*, was located on OAR12. This SNP had a number of candidate genes identified, but protein kinase C zeta (*PRKCZ*) is of specific interest. The *PRKCZ* gene synthesises an enzyme which belongs to the PKC family of serine/threonine kinases. These kinases are involved in many different cellular process such as positive regulation of cell proliferation (GO: 0008284), cell surface receptor signalling pathways (GO: 0007166) and negative regulation of protein-containing complex assemblies (GO: 0031333). Various studies within humans have found prior associations between protein kinase C’s and hair follicular growth and structure (i.e., length and elongation) [44,46].

*TMTC3* (transmembrane O-mannosyltransferase targeting cadherins 3) is associated with the 8-month fibre diameter phenotype and is a gene which encodes a protein belonging to the transmembrane and tetratricopeptide repeat-containing protein family. Identified biological processes of this protein include protein glycosylation (GO: 0006486), protein O-linked mannosylation (GO: 0035269) and positive regulation of proteasomal protein catabolic processes (GO: 1901800). Studies have reported that the TMTC gene family is associated with hair colour [47,48]. More specifically, the study conducted in Rhode Island Red Chicks showed that the gene of interest was associated with feather colour. Along with *TMTC3*, the *SLC45A1* (Solute Carrier Family 45 Member 1) gene, which encodes a protein from the glycoside-pentoside-hexuronide cation symporter transporter family, was identified as a candidate gene during the association of staple length at 8 months; notably, this gene has also been associated with hair colour in humans [48]. Considering the current and previous studies, it can suggested that the *TMTC3* and *SLC45A1* genes are involved in the mechanisms that affect colour, formation (i.e., diameter) and metabolism of hair and feathers in mammals and poultry.

## 5. Conclusions

In the present study, heritability estimates of four wool qualities and one wool production trait were found to be moderate to high. Subsequent GWAS identified one genome-wide and six chromosome-wide significant SNPs for wool diameter and length traits. Unfortunately, there were not any significant SNPs found for yearling greasy fleece weight. Within significant SNPs, there were several genes suggested as candidates for further analysis; accordingly, these were *TRIM2, MND1, TLR2, RNF175, CEP290, TMTC3, RERE, SLC45A1, SOX2, MORN1, SKI, FAAP20, PRKCZ, GABRD, CFAP74, CALML6* and *TMEM52*. Of specific interest are *CEP290*, *PRKCZ*, *TMTC3* and *SLC45A1* genes, which have a strong association to molecular and biological processes involved in the hair follicle development of sheep. This study reveals certain functional information about the genetics of wool quality and production traits. The current study promotes future plans to increase wool quality and yield within the Akkaraman breed through the design of efficient breeding programmes. However, more studies are required to fully understand the genetics of wool traits. This study contained a relatively low sample size, especially when considering the number of measurements in yearling animals. It is important for the improvement of the breed to carry out future studies on the genetics of these traits with higher sample sizes and with increased wool quality and production traits.

## Figures and Tables

**Figure 1 genes-14-00713-f001:**
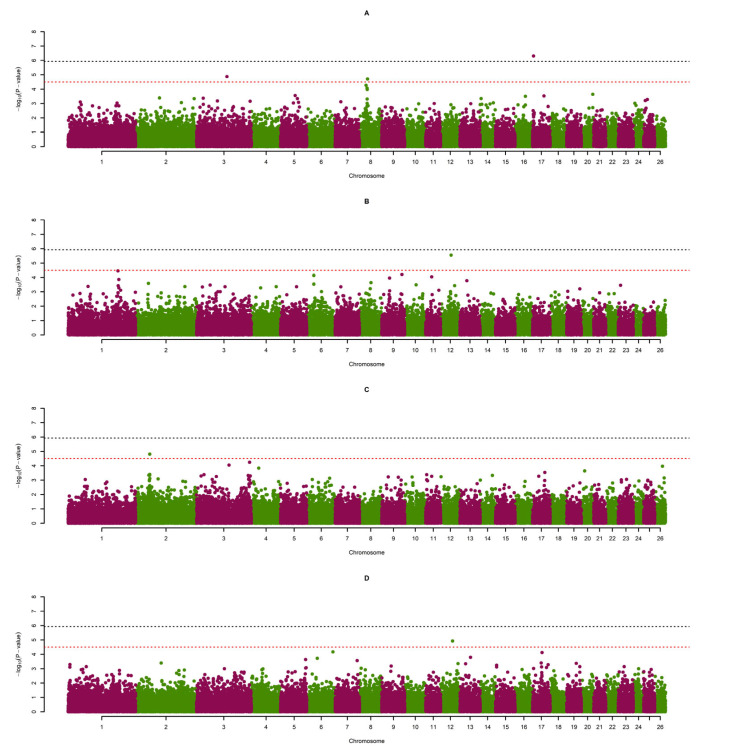
Manhattan plots of four wool traits. A black dashed line indicates genome-wide significance and a red dashed line indicates chromosome-wide significance. Fibre diameter at 8 months of age (FD) (**A**), staple length at 8 months of age (SL) (**B**), yearling fibre diameter (YFD) (**C**) and yearling staple length (YSL) (**D**) are the depicted traits of interest.

**Figure 2 genes-14-00713-f002:**
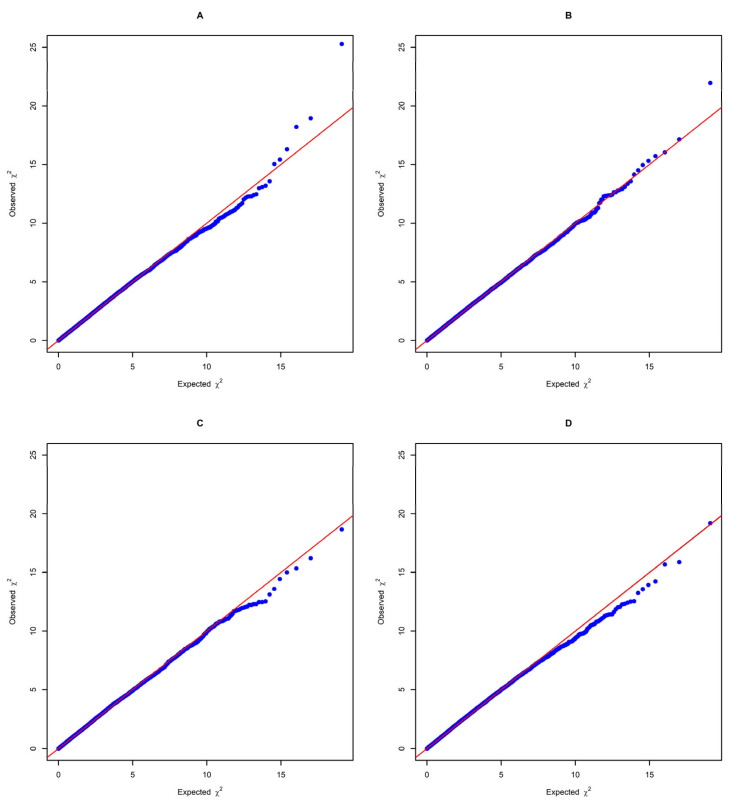
(**A**–**D**) Quantile–quantile (Q–Q) plots of genome-wide association studies (GWAS) for wool traits.

**Table 1 genes-14-00713-t001:** Descriptive statistics of wool traits.

Traits	Abr. ^1^	N ^2^	Mean ± SE ^3^	Min.	Max.	CV
**Fibre diameter at 8 months old**	FD	410	23.17 ± 0.13	16.43	31.49	0.12
**Staple length at 8 months old**	SL	426	34.50 ± 0.41	15.00	55.00	0.24
**Yearling fibre diameter**	YFD	78	22.73 ± 0.25	20.14	33.80	0.09
**Yearling staple length**	YSL	78	73.91 ± 1.96	35.00	120.00	0.23
**Yearling greasy fleece weight**	YGFW	78	2.43 ± 0.06	1.18	4.07	0.24

^1^ Abbreviation, ^2^ number of animals measured, ^3^ standard error.

**Table 2 genes-14-00713-t002:** Variance components and genomic heritability (h^2^) estimates for traits of focus.

Traits	Additive Genetic Variance ± SE	Residual Variance ± SE	h^2^ ± SE
**FD**	2.61 ± 0.97	3.72 ± 0.82	0.41 ± 0.13
**SL**	20.18 ± 10.13	60.50 ± 9.67	0.25 ± 0.12
**YFD**	0.23 ± 0.39	0.78 ± 0.38	0.22 ± 0.37
**YSL**	201.41 ± 154.29	117.88 ± 131.43	0.63 ± 0.43
**YGFW**	0.17 ± 0.15	0.18 ± 0.14	0.48 ± 0.40

h^2^ is genomic heritability and SE is standard error.

**Table 3 genes-14-00713-t003:** Significant SNPs associated with wool traits.

Traits	SNP Name	Chr.	Position (bp) ^1^	Alleles	*p*-Value	Significance Level	Associated Genes
**FD**	*rs399017576*	17	3977229	T/C	1.03 × 10^−7^	GW	*TRIM2, MND1, TLR2, RNF175, LOC101122892, LOC106991694*
**FD**	*rs416693666*	3	124062538	G/A	1.34 × 10^−5^	CW	*CEP290, TMTC3*
**FD**	*rs402446375*	8	32939307	C/T	1.97 × 10^−5^	CW	*-*
**SL**	*rs399379059*	12	43077532	T/C	1.12 × 10^−5^	CW	*RERE, SLC45A1, LOC101118971, LOC105609137*
**SL**	*rs420575132*	1	203515469	G/A	2.30 × 10^−5^	CW	*SOX2, LOC105603067*
**YFD**	*rs427654794*	2	57719968	T/C	1.55 × 10^−5^	CW	*-*
**YSL**	*rs419440256*	12	48639405	C/T	1.17 × 10^−5^	CW	*MORN1, SKI, FAAP20, PRKCZ, GABRD, CFAP74, CALML6, TMEM52, LOC106991467, LOC106991455, LOC105616534, LOC105609719*

^1^ SNP position based on OAR_v4.0; significance level is designated as either genome-wide (GW) or chromosome-wide (CW).

## Data Availability

The data presented in this study are available upon reasonable request from the corresponding author. The data are not publicly available due to the legal restriction of data deposition regarding indigenous breeds.

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
