# Peer review of "Genome-Wide Scan of Wool Production Traits in Akkaraman Sheep"

_genes, 2023, doi:10.3390/genes14030713_

Round 1

Reviewer 1 Report

Current work has focused on estimating heritability of traits from genomic data and used GWAS to screen relevant genomic regions for wool fibers and fleece characteristics. The authors reported that the heritability of wool fibre diameter, staple length, and yearling greasy fleece weight were estimated to be moderate to high. Moreover, in GWAS, significant SNPs of one genome-wide and six chromosome- wide for diameter and length traits were found. The findings provide a reference for the study of wool production traits of Akkaraman sheep. I would like to recommend to accept it after minor revision.

Specific comments:

1. Why were 8 months and 18 months sheep wool samples taken for subsequent analysis? And why were only 78 samples collected from sheep at the aged of 18 months-old?

2. The conditions of quality control should be detailed in the method.

3. Why is a measurement below or above 3 standard deviations considered an outlier?

4. Line 191-192: Please briefly describe how to convert non-normally distributed data into a normal distribution.

5. Figure 1C and D show that most points are below the diagonal line, indicating that the observed P value of most points is lower than the expected value. Why is this happening? Have you considered whether the model you choose is appropriate? Please explain in the discussion.

6. In Figure 2, label the locus above the threshold line corresponding to the gene name.

7. The discussion of the manuscript is a little bloated, so it is suggested to condense appropriately.

Author Response

REVIEWER 1

Dear Editor/s and Reviewers, thank you so much for providing revisions and guiding us through the process. We have implemented various changes and provided responses considering the comments of the reviewers. From our perspective, the comments, questions and suggestions of the reviewers were very informative and helpful for improving our manuscript at its current version. Therefore, we also would like to thank to the reviewers who profoundly contributed to our manuscript. Both reviewers suggested revisions in terms of the language of the manuscript. Therefore, a comprehensive language revision has been implemented by a native speaker. Other suggestions and relevant responses are given below as per the Reviewer.

  1. Why were 8 months and 18 months sheep wool samples taken for subsequent analysis? And why were only 78 samples collected from sheep at the aged of 18 months-old?

The animals of the study belong to three private large scale commercial herds. They were genotyped at around 6 months of age for the genetic improvement purposes of several other traits. 2 months after the genotyping (at around 8 months of old) due to the herd management practices, we have gained access to the genotyped animals with permission of collecting wool samples of those 426 animals. Later, for the second set of samples, 18 months of age was the first routine shearing period of those commercial farms. Only 78 out of those 426 animals that were initially phenotyped for wool characteristics were observed to be left in the farms for further sampling.

  1. The conditions of quality control should be detailed in the method.

Briefly, SNPs with call rate below 95 %, minor allele frequency lower than 0.05 and out of Hardy-Weinberg Equilibrium (p= 0.05/49,931) were removed from the analysis. Additionally, those SNPs that are mapped on the sex-linked chromosomes, mitochondrial chromosomes as well as the SNPs with unknown position were also removed from further analysis. On the other hand, animals were removed from the analysis if samples have call rate below 90 % and pairwise IBS rate higher than 95%. 

Following the suggestion of the reviewer, the same explanation provided above is also given in the “2.2. Genotype and quality control” section of the Material & Methods. 

  1. Why is a measurement below or above 3 standard deviations considered an outlier?

Considering, the Standard Normal Distribution’s bell-shaped curve, ± 3 Standard Deviation from the mean basically covers the 99.9 % of the data. Therefore, anything exceeding this can be cast out as for potentially being outlier due to measurement errors. It is a standard method of outlier removal in statistics. 

  1. Line 191-192: Please briefly describe how to convert non-normally distributed data into a normal distribution.

Yearling fibre diameter (YFD) was exposed to log transformation prior to any further analysis. It is basically to change the scale of the data into log10 base, which instantly removed the skewedness of the data and brought to a near normal distribution.  

As suggested by the reviewer, similar explanation was also inserted to the manuscript.

  1. Figure 1C and D show that most points are below the diagonal line, indicating that the observed P value of most points is lower than the expected value. Why is this happening? Have you considered whether the model you choose is appropriate? Please explain in the discussion.

The statistical model used for GWAS and further imposed ‘Genomic Control’ of the test statistics to correct for inflation are known to be conservative approaches which allow us avoid any false positive GWAS hits to prevent misleading results as also detailed in the last paragraph of “2.3. Statistical Analysis” of the Material and Methods. Overall, the used methods appear to be successful considering the Q-Q plots of all the analysed traits as there is no major systematic inflation of the test statistics. The special case of the YFD and YSL (i.e., the slight deflation of the test statistics) might be sourced by the very highly polygenic nature of the traits, which is also supported by the no association of any major SNPs regarding the traits of interest (Yang et al., 2011).

As pointed by the reviewer, a similar expression was also inserted to the end of the first paragraph of the “4.2. Genome wide association studies (GWAS)” in the Discussion.

  1. In Figure 2, label the locus above the threshold line corresponding to the gene name.

As the locus above the threshold line is not exactly a properly characterised gene, but an uncharacterized region, we thought it would be best not provide a single specific name on the Figure, as it could be misleading, but to suggest surrounding genes with a Table (i.e., see Table 3).

  1. The discussion of the manuscript is a little bloated, so it is suggested to condense appropriately.

We have specifically re-handled the discussion as suggested by the reviewer, which resulted in a more shrunk and compressed section. 

Reviewer 2 Report

Comments 

1. Is the Akkaraman breed coarse-, semi-coarse or semi-fine wool breed? This information should be provided in the Introduction.  

2. The aim of the study as well as conclusion are vague in the present form. As follows from the Introduction, the studied breed is dual purpose breed that produce meat and milk and has not been selected for wool production. Why this study is relevant? Is there national or regional program to improve the fleece related traits of this definite breed? Are there any specific products which are made of wool of Akkaraman breed? Please specify these points in the Introduction or Discussion sections.

3. The photograph of Akkaraman breed will be beneficial as well photographs of the fleece (or for example, the products made of wool of this breed if there are any) and support the reasoning of the manuscript.

4. How many females and males were among studied samples?

5. Please check the numeration of the tables (Table 3 on p 7 is marked as Table 2)

6. The Discussion related to the identified genes should be improved. Were these genes found in other sheep or goat breeds? Besides only few genes from those, which were identified in the study, are discussed. 

Author Response

REVIEWER 2

Dear Editor/s and Reviewers, thank you so much for providing revisions and guiding us through the process. We have implemented various changes and provided responses considering the comments of the reviewers. From our perspective, the comments, questions and suggestions of the reviewers were very informative and helpful for improving our manuscript at its current version. Therefore, we also would like to thank to the reviewers who profoundly contributed to our manuscript. Both reviewers suggested revisions in terms of the language of the manuscript. Therefore, a comprehensive language revision has been implemented by a native speaker. Other suggestions and relevant responses are given below as per the Reviewer.

  1. Is the Akkaraman breed coarse-, semi-coarse or semi-fine wool breed? This information should be provided in the Introduction.

Several studies suggested that Akkaraman sheep have a coarse- and carpet-type wool (Yalcin, 1986; Kaymakci et al., 2001). 

Similar expression inserted to the second paragraph of Introduction section, as requested by the reviewer.

  1. The aim of the study as well as conclusion are vague in the present form. As follows from the Introduction, the studied breed is dual purpose breed that produce meat and milk and has not been selected for wool production. Why this study is relevant? Is there national or regional program to improve the fleece related traits of this definite breed? Are there any specific products which are made of wool of Akkaraman breed? Please specify these points in the Introduction or Discussion sections.

As previously suggested, Akkaraman wool has long known to have coarse- carpet-type wool. Most of the wool produced by Akkaraman sheep has been used for blanket, rug and carpet production since a couple of historical studies suggest that Akkaraman sheep have low quality wool. However, these findings were based only on a few studies implemented decades ago. Therefore, no known selection were applied on the wool characteristics of Akkaraman sheep. However, in our study, we found that fibre diameter of Akkaraman wool is not that far from those mainstream sheep that are known for fine-wool production such as Merino. Therefore, with this study, we aimed to unravel 1) Overall wool characteristics of Akkaraman in comparison with those mainstream fine-wool sheep 2) Investigate the genetic background of those wool characteristics in Akkaraman sheep, so that in need of a genetic selection programme for wool characteristics, enhanced methods such as marker-assisted selection could be utilized for faster genetic improvement.

Upon the request of the reviewer, similar sentences were inserted as the first paragraph of Discussion section.

  1. The photograph of Akkaraman breed will be beneficial as well photographs of the fleece (or for example, the products made of wool of this breed if there are any) and support the reasoning of the manuscript.

As suggested by reviewer, the representative photos of Akkaraman sheep and its wool sample were taken by the authors and given as the Supplementary Figures 1 and 2.

  1. How many females and males were among studied samples?

As suggested by the reviewer, it was added to the related part of the Material and method (2.1. Animals and phenotypes).

  1. Please check the numeration of the tables (Table 3 on p7 is marked as Table 2)

It was corrected as requested by the reviewer.  

  1. The Discussion related to the identified genes should be improved. Were these genes found in other sheep or goat breeds? Besides only few genes from those, which were identified in the study, are discussed.

The reviewer 1 suggested the Discussion part of the manuscript to be shortened. We have specifically checked for whether the suggested genes were previously associated with wool, mohair, fleece and hair characteristics in sheep, goats and humans as well as other mammals such as mice. This method of reporting was chosen to keep the Discussion more focused, tidy and compact, we only reported the most biologically annotated genes in terms of these characteristics in all of the above mention species.  

Round 2

Reviewer 2 Report

It would be nice moving photographs of sheep from Supplementary Figures to the Main text.